When aggressiveness could be too risky: linking personality traits and predator response in superb fairy-wrens

Bilby Jack 1
Colombelli-Négrel Diane diane.colombelli-negrel@flinders.edu.au 1
Katsis Andrew C. 1
Kleindorfer Sonia 1 2
1 College of Science and Engineering, Flinders University , Bedford Park , South Australia , Australia
2 Konrad Lorenz Research Center for Behavior and Cognition & Department of Behavioral and Cognitive Biology, University of Vienna , Vienna , Austria
Hedrick Brandon
Electronic publication date: 2022 Sep 28
Publication date: 2022
Volume: 10
Electronic Location ID: e14011
Received 2022 Jan 21; Accepted 2022 Aug 15
Copyright: ©2022 Bilby et al.
Copyright year: 2022
Copyright holder: Bilby et al.
License: This is an open access article distributed under the terms of the Creative Commons Attribution License, which permits unrestricted use, distribution, reproduction and adaptation in any medium and for any purpose provided that it is properly attributed. For attribution, the original author(s), title, publication source (PeerJ) and either DOI or URL of the article must be cited.
License URL: https://creativecommons.org/licenses/by/4.0/

Keywords: Personality, Anti-predator behaviour, Risk-taking, Malurus

Funding: Australian Research Council DP190102894 The project was funded by the Australian Research Council (DP190102894). The funders had no role in study design, data collection and analysis, decision to publish, or preparation of the manuscript.

==============================
Personality syndromes in animals may have adaptive benefits for survival. For example, while engaging in predator deterrence, reactive individuals tend to prioritise their own survival, while proactive individuals engage in riskier behaviours. Studies linking animal personality measured in captivity with individual fitness or behaviours in the wild are sparse, which is a gap in knowledge this study aims to address. We used playback experiments in superb fairy-wrens (Malurus cyaneus), a common Australian songbird with a cooperative breeding system, to assess whether three personality traits measured during short-term captivity correlated with behavioural responses in the wild to a perceived nest and adult predator, the grey currawong (Strepera versicolor). We used three standard measures of personality in birds: struggle responses to human handling (boldness), exploration during a novel environment test, and aggressiveness during a mirror presentation. Superb fairy-wrens showed a significantly stronger response to the predator playback than to the control (willie wagtail, Rhipidura leucophrys) playback, suggesting that they recognised the predator playback as a threat without any accompanying visual stimulus. Birds that attacked their mirror image during the mirror presentation and those that spent a moderate amount of time close to the mirror responded more strongly to predator playback (by approaching the speaker faster and closer, spending more time near the speaker, and being more likely to alarm call) compared to those with low aggressiveness or those that spent very short or long durations close to the mirror. Neither boldness nor exploration in the novel environment test predicted playback response. Our results align with a growing number of studies across species showing the importance of animal personalities as factors for fitness and survival.

Introduction

In animals, individuals with distinct personality traits can engage in different risk-taking strategies (Jones & Godin, 2010; Quinn & Cresswell, 2005; Van Oers et al., 2004). Personality, also known as temperament or coping strategy, refers to between-individual differences in behavioural traits that are consistent over time and across contexts (Bókony et al., 2012; Hall et al., 2015; Réale et al., 2007). Personality traits have been demonstrated in a number of taxa (Bell, Hankison & Laskowski, 2009), may be partly heritable (Dingemanse et al., 2002), and can remain consistent for a significant portion of an individual’s lifespan (Hall et al., 2015; Wuerz & Krüger, 2015). Personality traits can be measured along five broad axes, which may correlate with each other to form a behavioural syndrome along the ‘proactive-reactive’ axis (Sih, Bell & Johnson, 2004): (a) the shyness-boldness axis (how an individual reacts during a risky situation, such as encounters with predators and humans); (b) the exploration-avoidance axis (how an individual reacts to a new situation, such as a novel environment, food, or object); (c) the low-high activity axis (the general level of activity of an individual in a non-risky and non-novel environment); (d) the low-high aggressiveness axis (an individual’s agonistic reaction towards conspecifics); and (e) the low-high sociability axis (how an individual reacts to the presence or absence of conspecifics when excluding aggressive behaviour, Réale et al., 2007; Table 1; Fig. 1). Most studies examining animal personality traits in different contexts have focused on the shyness-boldness and exploration-avoidance axes (e.g., Amy et al., 2010; Jacobs et al., 2014; Snijders et al., 2015; van Asten, Hall & Mulder, 2016) or how one personality trait may relate to other behaviours or personality traits (Duckworth, 2006; Hall et al., 2017; Hollander et al., 2008). Although individual differences in personality traits have been shown to correlate with survival and fitness (Dingemanse et al., 2004; Dingemanse & Réale, 2005; Smith & Blumstein, 2008), studies explicitly linking animal personality measured in captivity with behaviours in the wild are sparse, and much less is known about how personality traits in one context relate to the same traits in other contexts (but see Cain et al., 2011; Huntingford, 1976; Witsenburg et al., 2010).

Table 1 Methodological framework correlating animal personality measured in captivity and in the wild (adapted from Réale et al. (2007)).

Personality traits and measurements used in this study are marked in bold.

Personality axis	Definition	Example tests in captivity	Example tests in the wild	
(A) Shyness-boldness	How an individual reacts during a risky situation, such as encounters with predators and humans	Response to human handling (‘back-test response’, ‘processing response’)	Flight initiation distance experiments; response to simulated predator	
(B) Exploration-avoidance	How an individual reacts to a new situation, such as a new habitat, food, or object	Cage novel environment test; presentation of novel object in cage	Open field test in a novel environment; presentation of novel object in the field	
(C) Low-High Activity	General level of activity of an individual in a non-risky and non-novel environment	Cage activity test	Open field test	
(D) Low-High Aggressiveness	An individual’s agonistic reaction to conspecifics	Mirror stimulation test	Territory defence experiments	
(E) Low-High Sociability	How an individual reacts to the presence or absence of conspecifics (excluding aggressive behaviour)	Separation test	Network analysis	

Figure 1 Hypotheses linking personality traits measured in captivity and in the wild.

Personality traits and hypotheses linked to this study are marked in bold.

In risky situations, such as encounters with predators, different personality types may have distinct adaptive benefits (Aplin et al., 2014; Coleman & Wilson, 1998; Hedrick & Riechert, 1993; Sih, Kats & Maurer, 2003; Verbeek, Drent & Wiepkema, 1994). Fast-exploring individuals often at the proactive end of the ‘proactive-reactive’ axis, where a ‘proactive’ bird is not just fast-exploring, but also bold, aggressive and active, (Sih, Bell & Johnson, 2004; Table 1; Fig. 1) tend to engage in riskier behaviours, such as investigating novel objects/environments (Huntingford, 1976; Kluen et al., 2012; Koolhaas et al., 2001; Verbeek, Drent & Wiepkema, 1994), acting conspicuously in the presence of a predator (Cole & Quinn, 2014; Greig, Spendel & Brandley, 2010; Langmore & Mulder, 1992; Sih, Kats & Maurer, 2003), and returning to typical behaviour more quickly after a threat has passed (Hedrick & Riechert, 1993; Verbeek, Drent & Wiepkema, 1994). In contrast, slow-exploring individuals (at the reactive end of the axis) tend to prioritise their own survival and are, therefore, more risk-averse (Coleman & Wilson, 1998; Hall et al., 2015; Quinn & Cresswell, 2005). Individuals’ ability to avoid potential predators may thus drive differential selection patterns amongst personality types (Jones & Godin, 2010; Quinn & Cresswell, 2005), especially if individuals have to trade off between predation risk and other important behaviours such as foraging (Jones & Godin, 2010; Sih, Bell & Johnson, 2004; Verbeek, Drent & Wiepkema, 1994). For instance, mole salamanders (Ambystoma barbouri) that were highly active foragers were also active in the presence of a perceived predator, making them more vulnerable to predation (Sih, Kats & Maurer, 2003). For this reason, risk-taking behaviours that enhance activity may be maladaptive when predation risk is high.

An individual’s sex, regardless of its personality, may also affect how it responds to potential predators (McQueen et al., 2017; Samia et al., 2015; Tanis, Bott & Gaston, 2018; Zelano, Tarvin & Pruett-Jones, 2001). Many bird species are sexually dimorphic, with males displaying more conspicuous plumage than females, which can lead to sex-specific differences in both predation risk and anti-predator behaviours (McQueen et al., 2017; Powolny et al., 2014; Ruiz-Rodríguez et al., 2013; Stuart-Fox et al., 2003, but see Cain et al., 2011). However, the relationship between conspicuousness or sex and anti-predator behaviours is not straightforward, simply because individuals also differ in other factors that influence risk-taking behaviours. For example, when early parental investment is higher for female birds (through the costs of egg laying and incubation), mothers may tolerate higher levels of risk when defending their investment and nest (Trivers, 1972). During the breeding season, females may also engage in riskier behaviours because they have higher caloric needs than males (Powolny et al., 2014). The interaction between personality, conspicuousness and sex is relatively unexplored, but, in general, conspicuous individuals tend to be more vigilant and risk-averse in the presence of predators than their less colourful conspecifics (Hart & Freed, 2005; Ibáñez, López & Martín, 2014; Journey et al., 2013; McQueen et al., 2017; Ortega, López & Martín, 2014).

In this study, we used wild superb fairy-wrens (Malurus cyaneus), a common Australian songbird with a cooperative breeding system, to assess whether sex and three personality traits (boldness, exploration, and aggressiveness), measured during short-term captivity, correlated with boldness in the wild (Table 1; Fig. 1). We assessed boldness in wild fairy-wrens by using vocal playback to simulate the presence of a local predator of nestlings and adults, the grey currawong (Strepera versicolor) (Colombelli-Négrel, Robertson & Kleindorfer, 2009a; Colombelli-Négrel, Robertson & Kleindorfer, 2010a; Greig, Spendel & Brandley, 2010). A previous study found that fast-exploring superb fairy-wrens (tested in a novel environment test) were less likely to be present in the wild one year after initial sampling (Hall et al., 2015). Because males and breeding females are highly philopatric in this species, the authors suggested that these disappearances were likely explained by increased mortality (Hall et al., 2015; see also Cockburn et al., 2008), due to fast-exploring individuals being less cautious and taking more risks (i.e., approaching faster and closer or resuming foraging sooner once the threat has supposedly passed) when faced with potential threats (Dingemanse & Réale, 2005; Hall et al., 2015; Jones & Godin, 2010). Here, we tested this hypothesis by broadcasting predator vocalisations close to the nests of fairy-wrens with known personalities and assessing each group member’s behavioural response. Based on Hall et al. (2015), we predicted that fast-exploring individuals would be more likely to show risky behaviours in the presence of a predator, thereby increasing their mortality risk. These behaviours may include responding more slowly to the predator playback or approaching closer to the speaker and staying longer in its proximity (see also Jones & Godin, 2010; Quinn & Cresswell, 2005) than slow-exploring individuals. We also predicted that males would be more risk-averse in response to the predator playback compared to females, due to the males’ more conspicuous colouration (Hart & Freed, 2005; Ibáñez, López & Martín, 2014; Journey et al., 2013; McQueen et al., 2017; Ortega, López & Martín, 2014).

Methods

Study site and species

We conducted our study using a wild population of superb fairy-wrens at Cleland Wildlife Park (34°58′S, 138°41′E), located 12 km southeast of Adelaide in the Mount Lofty Ranges, South Australia. The park is a mosaic habitat of stringybark forest, including blue gum (Eucalyptus leucoxylon), manna gum (Eucalyptus viminalis), open grassland, and a complex understory of trees and shrubs (described in Colombelli-Négrel & Evans, 2017). All birds were colour-banded, and all territories have been monitored since 2010. Cleland Wildlife Park hosts approximately 27 territories with 60+ adult fairy-wrens. We measured personality traits between November 2019 and January 2020 or between September 2020 and January 2021 and conducted all playback experiments between October and November 2020. These periods corresponded with the Austral spring and summer and with the peak of the fairy-wrens’ breeding season.

Superb fairy-wrens are small, insectivorous passerines endemic to south-eastern Australia (Rowley & Russell, 1997). Adult birds are socially monogamous and sexually promiscuous, with about 95% of all nests containing at least one egg sired by another male (Colombelli-Négrel, Schlotfeldt & Kleindorfer, 2009b; Mulder et al., 1994). They are cooperative breeders, where young males (and more rarely females) can remain in their own territory to provide help to the dominant pair (Cockburn et al., 2008; Kleindorfer et al., 2013b; Margraf & Cockburn, 2013; Mulder et al., 1994). At our study site, group size ranged from two to six adults (mean ± SE number of adults = 3 ± 0.22), and helpers could be either male or female (with 20% of the monitored groups having at least one female helper). During the breeding season (approx. September to January), females build a dome-shaped nest hidden in dense vegetation and incubate the eggs alone (Colombelli-Négrel & Kleindorfer, 2009). Females lay two to four eggs (typically three) and incubate them for ∼14 days (Rowley & Russell, 1997). All group members help to feed and defend the offspring (Colombelli-Négrel et al., 2010b). Young birds fledge from 10–14 days old (Rowley & Russell, 1997). In other parks within the Mount Lofty Ranges, annual nest predation of superb fairy-wren nests varies from 34 to 83% (Kleindorfer et al., 2014) while annual nest predation at Cleland Wildlife Park varies from 25 to 78% (Colombelli-Négrel & Kleindorfer, pers. obs. 2008–2021). Nest predators include grey currawongs, house mice (Mus musculus) and bush rats (Rattus fuscipes) (Colombelli-Négrel, Robertson & Kleindorfer, 2009a). In this study, we focused on superb fairy-wrens’ response to the grey currawong, an avian predator that poses a threat to both adults and nestlings (Colombelli-Négrel, Robertson & Kleindorfer, 2009a; Magrath, Pitcher & Gardner, 2009). When confronted with an aerial predator, such as the grey currawong, superb fairy-wrens respond with various vocalisations, including: (1) a terrestrial alarm call (also referred to as “chits” call or “mobbing alarm call”), used when an aerial predator is on the ground or perched (Colombelli-Négrel, Robertson & Kleindorfer, 2010a; Rowley & Russell, 1997); (2) an aerial alarm call (“flee” call), used when an aerial predator is flying overhead (Rowley & Russell, 1997); (3) a trill song (or type II song), produced by males upon hearing loud avian predator calls (Greig & Pruett-Jones, 2008; Langmore & Mulder, 1992; Zelano, Tarvin & Pruett-Jones, 2001); and (4) an alarm song (or type III song), produced under immediate threat of predation (Colombelli-Négrel, Robertson & Kleindorfer, 2011).

Personality assays

To assess individual personality traits, adult superb fairy-wrens were captured using mist-nests and taken to a nearby banding station. We first assessed their boldness phenotype (personality trait 1) by measuring the birds’ response to human handling, following procedures modified from Brommer & Kluen (2012). Individual responses to human handling have been identified as repeatable behavioural traits in other species, including in adult and nestling birds (Hall et al., 2015; Kluen, Siitari & Brommer, 2014). During handling, we recorded two behavioural variables. First, we tilted the bird onto its back in the bander’s grip and counted the number of individual struggles over a 30 s period (‘back-test response’, also called ‘docility’) (Brommer & Kluen, 2012; Hall et al., 2015; Hessing et al., 1993) (Fig. 2A). Second, we noted whether the subject struggled while recording five morphological measurements, and then assigned it a discrete ordinal score from 0 (did not struggle during any measurement procedure) to 5 (struggled during all five measurement procedures). The morphological measurements were: (1) tarsus length (to the nearest 0.1 mm, using a sliding calliper), (2) head-bill length (nearest 0.1 mm, sliding calliper), (3) tail length (nearest one mm, ruler), (4) wing length (nearest one mm, butt-ended ruler), and (5) mass (nearest 0.01 g, electronic scale). We named this behavioural variable ‘processing response’ (also called ‘handling aggression’; see Brommer & Kluen, 2012). If not already banded, the bird was fitted with a numbered aluminium leg-band issued by the Australian Bird and Bat Banding Scheme and a unique combination of three coloured bands to allow visual identification in the field.

Figure 2 Photographs of the behavioural assays used to measure personality traits in this study.

(A) A fairy-wren tilted onto its back during the back-test handling assay, and (B) the flight cage used for the novel environment test and mirror stimulation test, with the mirror revealed and a fairy-wren observing its mirror image.

All captured adults were then transferred in a cotton bag to an onsite building, where we quantified their exploration phenotype (personality trait 2) during a novel environment test (Fig. 2B) and their aggressiveness phenotype (personality trait 3) using a mirror stimulation test. Each subject was placed alone in a small wooden release box (170 × 115 × 90 mm in size) and allowed 5 min to acclimate. The door to the release box was then opened, allowing the bird to enter the novel environment: a metal flight cage (700 × 450 × 450 mm) with three wooden perches (Fig. 2B). The cage was covered with an opaque fabric on all but one side, visually isolating the birds from its surroundings. On the remaining side, we had already placed a camera (GoPro Hero7 Black, GoPro, Inc., San Mateo, CA, USA) to observe the subject’s movements. The novel environment was divided into 13 distinct sectors that the subject could visit: three perches, four floor quadrats, four cage walls, the cage ceiling, and the release box. For each subject, we recorded (1) ‘activity’ (the total number of sectors that the bird visited in the 5 min following emergence, including repeat visits) and (2) ‘sector visitation’ (the number of unique sectors visited over the same period, also referred to as ‘exploration’; see Hall et al., 2015). After 5 min, we remotely raised a curtain at one end of the cage to reveal a mirror (300 × 400 mm), which commenced the mirror stimulation test (Fig. 2B). Over the next 3 min, we then recorded two measures of aggressiveness: (1) ‘mirror attacks’ (the number of times the bird contacted the mirror) and (2) ‘time close to mirror’ [the total time (in seconds) the bird spent close to the mirror (i.e., in the three nearest sectors excluding time spent attacking the mirror)]. Video recordings of these assays were scored using the software Solomon Coder v. beta 19.08.02. To avoid any observer bias in the testing procedure, the novel environment and mirror stimulation tests were conducted and scored by a single experimenter (ACK). Some birds were tested multiple times; however, for this study, we only analysed performance in their first trial, when the novel environment and mirror presentation were still truly novel. In our study population, responses to the novel environment were significantly repeatable (activity: adjusted R = 0.303, P = 0.005; sector visitation: adjusted R = 0.311, P = 0.005, comparable to the adjusted repeatability of 0.37 previously found in superb fairy-wrens by Hall et al. (2015)) and the repeatability of aggressiveness (likelihood of attacking the mirror) during the mirror-stimulation test was marginally non-significant (adjusted R = 0.152, P = 0.062), based on N = 226 trials by 150 birds (Katsis, Colombelli-Négrel & Kleindorfer, pers. obs. 2019–2021).

Playback responses

From October to November 2020, we used playback experiments to investigate the relationship between personality traits measured in captivity and predator response in the wild. We used two playbacks for stimuli: (1) the flight call (i.e., produced when the predator is in flight) of a local fairy-wren predator, the grey currawong, and (2) the song of a willie wagtail (Rhipidura leucophrys), a local bird with no antagonistic interactions with superb fairy-wrens and used as our control. We sourced the willie wagtail playback songs from non-local individuals from five recordings (tracks: XC608504; XC596007; XC578439; XC233822; XC151592) from the xeno-canto database (https://www.xeno-canto.org/). We sourced the grey currawong playback calls from recordings made in 2009 at Newland Head Conservation Park (35°37′S, 138°29′E) and Scott Creek Conservation Park (35°05′S, 138°41′E). These calls were recorded as broadcast wave files (48 kHz sampling rate, 24-bit depth) using a Telinga parabolic microphone (Telinga Microphones, Sweden) connected to a portable Sound Devices 722 digital audio recorder (Sound Devices LLC, Reedsburg, WA, USA). We used five recordings for the grey currawong and five recordings for the willie wagtail to create five grey currawong playback tracks and five willie wagtail playback tracks. We used three different calls/songs from the same recording to create each track. We prepared all tracks using Amadeus Pro 2.2 (Hairersoft Inc., Switzerland) and exported all files as uncompressed 16-bit Broadcast wave files (.wav) to an Apple iPod (Apple Inc., Cupertino, CA, USA). Each playback track was 9 min long, comprising 3 min of silence (pre-playback), followed by 3 min of playback (playback) and another 3 min of silence (post-playback). The playback phase was structured as follows: 1 min of calls (one call/song every 10 s; total six calls/songs per minute), 1 min of silence, and a repetition of the 1 min of calls/songs. Each track was used on average 2.8 ±0.39 times (range 1–5 times).

We tested all birds within a group simultaneously during the early incubation period (days 2–7 of incubation), between 9–11am, to coincide with their most active foraging period (Colombelli-Négrel & Evans, 2017). We targeted the first nest of the season during early incubation, to reduce the likelihood that previous nest predation attempts within the same breeding season could influence the response to playback. All subjects were tested with both the predator and control playbacks on the same day, separated by 1–2 h and in a randomised order. Upon entering a fairy-wren territory, we placed a portable speaker (Sony XB12, Sony Australia Limited; frequency response 20 Hz–20 kHz) within 3 m of the nest, connected via Bluetooth to an Apple iPod. We never started playback until females were off the nest and at least one group member was observed within a 20 m radius of the speaker. Once we ensured that at least one bird was present, we started the playback track immediately. All playback tracks were played at ∼90 dB SPL (measured 1 m in front of the speaker). During the 9-minute playback trial, two observers posted within the territory (∼10 m from the speaker) continually observed the subjects’ behavioural responses using binoculars and narrated them into a directional microphone (Sennheiser ME67; Sennheiser electronic GmbH & Co., Wedemark, Germany) connected to an audio recorder (Zoom H6 recorder; Zoom, San Jose, CA, USA). For each subject, we recorded its: (1) minimum distance (m) from the speaker, (2) latency (s) to respond to the playback, and (3) time spent (s) within 5 m of the speaker. We also recorded the number of terrestrial alarm calls produced by subjects during the pre-playback period (no aerial alarm calls, Type II or Type III songs were produced during the 9 min of observation). During playback, it was not possible to keep track of the numbers of terrestrial alarm calls for each bird, due to the high number of group members calling at once. Instead, we noted whether each bird alarm called as a binary variable (yes/no). All individuals returned to foraging behaviours very quickly (pers. obs., not measured in this study); thus, we did not include the post-playback period in our analyses.

Ethical Note

All birds were banded under permit from the Australian Bird and Bat Banding Scheme (banding authority number 2601). Research was approved by the Flinders University Animal Welfare Committee (permit E480-19). We conducted fieldwork at Cleland Wildlife Park under permit Z24699 (approved by the South Australian Department of Environment and Water).

Statistical analysis

We analysed all data using SPSS 25.0 for Windows (SPSS Inc., Chicago, IL, USA). Playback data (minimum distance, latency to respond, time within 5 m, and number of terrestrial alarm calls produced during the pre-playback period) did not satisfy conditions of normality (Shapiro–Wilk tests: all P < 0.0001). We used Mann–Whitney tests to compare playback response to the predator (grey currawong) playback versus the control (willie wagtail) playback. As predicted, subjects did not noticeably respond to the control playback (see Results), so we focused exclusively on the predator playback for subsequent analyses. We used principal component analysis (PCA) to reduce several sets of correlated behavioural variables and playback responses to a smaller number of uncorrelated principal components (PCs). The first PCA included ‘back-test response’ and ‘processing response’ and produced a single component (PC_Handling) with an eigenvalue of 1.21 that accounted for 60% of variance (Table 2A). The second PCA included ‘sector visitation’ and ‘activity’ and produced a single component (PC_Exploration) with an eigenvalue of 1.53 that explained 77% of variance (Table 2B). The third PCA included ‘latency’, ‘minimum distance’, ‘time within 5 m’ and ‘alarm called’ and produced a single component (PC_Playback) with an eigenvalue of 2.65 that accounted for 66% of variance (Table 2C). Higher PC scores indicated a stronger response (i.e., more struggles for PC_Handling, more sectors visited for PC_Exploration, and a shorter latency to respond, a shorter minimum distance, more time spent close the speaker, and production of alarm calls for PC_Playback). Many birds (45% of 40 birds) did not attack the mirror at all during the mirror stimulation test; therefore, we converted ‘mirror attacks’ to a binary variable (attacked the mirror, did not attack the mirror). We used Spearman correlations to test for a correlation between aggressiveness (‘time close to mirror’), boldness (PC_Handling) and exploration (PC_Exploration) and Mann–Whitney U tests to test for a correlation between our binary measure of aggressiveness (‘mirror attacks’), ‘time close to mirror’, boldness (PC_Handling) and exploration (PC_Exploration).

Table 2 Factor loadings from principal component analysis of superb fairy-wren (A) boldness (response to human handling), (B) exploration (exploration behaviour during a novel environment test), and (C) playback response (latency, minimum distance, time within 5 m, alarmed called) (N = 40 individuals).

Higher scores indicated a stronger response: more struggles for PC_Handling, more sectors visited in the novel environment for PC_Exploration, and a shorter latency to respond, a shorter minimum distance, more time spent near the speaker, and production of alarm calls for PC_Playback. The eigenvalues and the percentage of the variance explained by each factor are presented in brackets.

(A) Boldness	PC_Handling (1.21; 60%)	
Back-test response	0.78	
Processing response	0.78	
(B) Exploration	PC_Exploration (1.53; 77%)	
Sector visitation	0.88	
Activity	0.88	
(C) Playback response	PC_Playback (2.65; 66%)	
Latency (s)	−0.84	
Min distance (m)	−0.91	
Time within 5 m	0.73	
Terrestrial alarm calls	0.77	

To test whether playback response (PC_Playback) correlated with our three personality traits (boldness, exploration, and aggressiveness) or sex, we used two generalised linear mixed models (GLMMs). Model 1 included boldness (PC_Handling), exploration (PC_Exploration), aggressiveness (‘mirror attacks’ as a binary variable), sex (male, female), ‘playback order’ (whether the predator playback was conducted first or second), and number of responders (the number of group members that responded to the playback) as fixed effects. Model 2 was identical to the above but replaced aggressiveness (‘mirror attacks’) with our second measure of aggressiveness (‘time close to mirror’ as a continuous and transformed into a quadratic variable, ‘time close to mirror’ × ‘time close to mirror’). We added number of responders to our analyses as fairy-wrens have been shown to adjust their response to predators or other threats depending on the presence of conspecifics (Teunissen et al., 2021; van Asten, Hall & Mulder, 2016). We analysed all birds together and did not separate helpers and dominant individuals. Both models had a Gaussian distribution with identity link and included ‘Territory ID’ (to account for variation among groups) as a random effect. The models did not reach convergence when ‘Playback ID’ was included; therefore, it was excluded as random effect. We present in supplementary material the results of Model 3, which was identical to the two models above but for the aggressiveness measure used ‘mirror attacks’ (as a continuous and transformed into a quadratic variable, ‘mirror attacks’ × ‘mirror attacks’).

Results

We analysed the predator playback responses of 40 birds (29 males and 11 females) with known personality across 15 territories. During the pre-playback period, there was no difference in the number of terrestrial alarm calls produced between the control and predator playback trials (Mann–Whitney: U = 780.50, N = 40, P = 0.57). As we predicted, individuals responded faster (U = 452.50, P < 0.0001) and were more likely to produce terrestrial alarm calls (U = 460.00, P <  0.0001) during the predator playback compared to the control playback. They also spent more time within 5 m of the speaker (U = 628.50, P = 0.04) in response to the predator playback, compared to the control. The minimum distance of approach did not differ between the two playback treatments (U = 607.00, P = 0.06). Among our personality traits measured in captivity, there was no significant correlation between boldness (PC_Handling) and exploration (PC_Exploration) (r2 = 0.02, P = 0.88), nor between boldness (PC_Handling) and aggressiveness (‘time close to mirror’: r2 = 0.06, P = 0.71; ‘mirror attacks’: U = 149, P = 0.19). There was a correlation between aggressiveness and exploration (PC_Exploration), with birds spending more time close to the mirror being more exploratory (‘time close to mirror’: r2 = 0.41, P = 0.01; Fig. 3A) and those that attacked the mirror tending to be more exploratory (U = 132, P = 0.07; Fig. 3B). Our two measures of aggressiveness were highly correlated, with birds that attacked the mirror many times also spending more time close to the mirror (U = 40.50, P < 0.0001).

Figure 3 Relationship between aggressiveness (‘mirror attacks’ and ‘time close to mirror’) and exploration (PC_Exploration) in superb fairy-wrens (N = 40).

The data are presented for (A) ‘mirror attacks’ (attacked the mirror, did not attack the mirror) and (B) ‘time close to mirror’ (total time in seconds the bird spent close to the mirror). Higher scores for exploration indicated more sectors visited in the novel environment. Horizontal lines within the boxes represent the means. The upper and lower limits of the boxes show the 75th and 25th percentiles, respectively. Black circles indicate outliers.

When considering only birds’ response to the predator playback, individuals that attacked the mirror during the mirror stimulation test showed a stronger response to the playback: they approached faster (mean approach time: 88 s versus 215 s), closer (mean minimum distance: 6.4 m versus 10.1 m), spent more time within 5 m of the speaker (mean time spent: 20 s versus 16 s), and were more likely to alarm call than those that did not attack the mirror (53% versus 50%) (Table 3; Fig. 4A). Birds that spent an intermediate amount of time near the mirror during the mirror stimulation test also responded more strongly to playback—they approached faster, closer, spent more time within 5 m of the speaker, and were more likely to alarm call—compared to birds at both extremes (Table 3; Fig. 4B). Boldness (PC_Handling), exploration (PC_Exploration), sex, playback order and the number of responders did not predict PC_Playback (Table 3). Territory ID did not significantly predict PC_Playback (Wald Z = 1.37; P = 0.17).

Table 3 Output from GLMMs testing whether personality traits measured during short-term captivity (boldness, exploration, and aggressiveness) correlated with behaviours in the wild in response to a simulated local predator (PC_Playback).

Model 1 included (1) boldness (PC_Handling), (2) exploration (PC_Exploration), (3) aggressiveness (‘mirror attacks’, as a binary variable), (4) sex (male, female), (5) playback order (whether the predator playback was conducted first or second), and (6) number of responders (the number of group members that responded to the playback) as fixed factors and ‘Territory ID’ as a random effect. Model 2 was identical to the above, except that aggressiveness (‘mirror attacks’) was replaced with aggressiveness (‘time close to mirror’ as a quadratic variable). Statistically significant (≤ 0.05) values are marked in bold (N = 40 individuals).

Fixed effects	Estimate	Std. error	t	P	
Model 1					
Intercept	−0.05	0.65	−0.07	0.95	
Boldness	0.09	0.15	0.59	0.56	
Exploration	−0.23	0.16	−1.43	0.16	
Mirror attacks (binary)	−0.82	0.37	−2.22	0.03	
Sex	−0.02	0.34	−0.06	0.95	
Playback order	0.40	0.45	0.90	0.37	
Number of responders	0.17	0.27	0.64	0.53	
Model 2					
Intercept	−0.76	0.79	−0.97	0.34	
Boldness	0.11	0.15	0.72	0.47	
Exploration	−0.34	0.19	−1.80	0.08	
Time close to mirror (continuous)	0.03	0.01	2.10	0.04	
Time close to mirror (quadratic)	<0.0001	<0.0001	−2.00	0.05	
Sex	−0.17	0.33	−0.52	0.61	
Playback order	0.10	0.48	0.21	0.84	
Number of responders	0.15	0.30	0.51	0.61	

Figure 4 Relationship between aggressiveness (‘mirror attacks’ and ‘time close to mirror’) and behavioural responses to predator playback (PC_Playback) in superb fairy-wrens (N = 40).

The data are presented for (A) ‘mirror attacks’ (attacked the mirror, did not attack the mirror) and (B) ‘time close to mirror’ (total time in seconds the bird spent close to the mirror). Higher scores for playback response (PC_Playback) indicated a stronger response: a shorter latency to respond, a shorter minimum distance, more time spent close the speaker, and production of alarm calls. Horizontal lines within the boxes represent the means. The upper and lower limits of the boxes show the 75th and 25th percentiles, respectively. Black circles indicate outliers.

Discussion

Previous research has shown that individuals with distinct personality traits differ in their risk-taking strategies (Jones & Godin, 2010; Quinn & Cresswell, 2005; Van Oers et al., 2004). Here, too, we found that an individual’s aggressiveness, measured as its response towards an apparent conspecific during short-term captivity, was associated with boldness and risk-taking in the wild. Specifically, superb fairy-wrens that attacked their mirror image and spent an intermediate period near the mirror during the mirror stimulation test responded most vigorously to predator playback. Conversely, exploration during a novel environment test and boldness (measured as response to human handling) did not predict an individual’s response to predator playback. These results suggest that specific personality traits, rather than an overall behavioural syndrome (where personality traits correlate with each other; Sih, Bell & Johnson, 2004), predict behavioural responses to a perceived predator in superb fairy-wrens.

Superb fairy-wrens that attacked their mirror image during the mirror stimulation test, and those that spent an intermediate duration near the mirror, responded most vigorously to the predator playback, approaching the speaker faster and closer, spending more time near the speaker, and being more likely to alarm call. This suggests that conspecific aggressiveness is associated with predator-response behaviours in this species, albeit in a non-linear fashion. Responding to a predator quickly and closely could reduce the risk of nest predation, by alerting other group members to the threat and facilitating the performance of distraction displays to draw the predators’ attention (Kleindorfer, Fessl & Hoi, 2005; LaBarge et al., 2021; Rowley, 1962; Zuberbühler, 2001). However, such a strong response would presumably also place the responders at greater risk of being predated themselves, making the net fitness benefits unclear. Aggressive phenotypes may be maintained in populations not because their predator responses are beneficial but because aggressiveness is associated with positive outcomes in other contexts: for example, during contests with conspecifics (Godin & Dugatkin, 1996; Hedrick & Riechert, 1993; Sih, Bell & Johnson, 2004) or during foraging, perhaps by helping individuals compete for shared food resources (Aplin et al., 2014; Coleman & Wilson, 1998; Hedrick & Riechert, 1993; Sih, Kats & Maurer, 2003). In the case of superb fairy-wrens, neighbouring groups probably compete with each other for the high-resource territories, and higher aggressiveness may be advantageous for these disputes (Cain & Langmore, 2016). The fitness benefits of aggressiveness are still to be fully investigated in superb fairy-wrens, although Hall et al. (2015) reported that fast-exploring birds (which also tended to be more aggressive) were less likely to be present in the wild one year after initial sampling, which suggests lower survivorship (although this remains to be fully tested). If aggressiveness is beneficial for competition with conspecifics, it may give individuals an adaptive benefit where predation risk is low (Hedrick & Riechert, 1993; Sih, Kats & Maurer, 2003). Conversely, where the risk of predation is high, more aggressive phenotypes that respond riskily to potential predators may be selected against. Unexpectedly, birds that spent the longest duration close to the mirror did not exhibit the strongest responses to predator playback in the wild. This could indicate that, beyond a certain threshold, very high scores in the mirror stimulation test no longer quantify a bird’s aggressiveness but perhaps also its intelligence (the ability to eventually recognise its mirror-image) or its perseveration (the tendency to perform recurring or redundant responses without appropriate stimulus; Vickery & Mason, 2005).

There was no significant relationship between the number of responders to the predator playback and the strength of individuals’ response. This contrasts with previous studies in fairy-wren species, where social context influenced individuals’ responses to a simulated predator or other potential threat (Teunissen et al., 2021; van Asten, Hall & Mulder, 2016). van Asten, Hall & Mulder (2016) found that fast-exploring superb fairy-wrens responded more strongly to a novel object than did slow-explorers, but only when multiple group members were present. It is possible that our study achieved different results due to the type of stimulus used to simulate a predator. As predatory birds will typically hunt in silence, it is possible that the perceived threat from our vocal playback trials was lower compared to what a physical model might produce (Abbey-Lee et al., 2016), which could in turn have influenced the relationship between the number of responders and playback response.

Fairy-wrens responded more strongly to the predator (grey currawong) playback than to the control (willie wagtail) playback. This indicates, consistent with other studies, that subjects recognised the playback track as a risk to the nest or to themselves without any accompanying visual stimuli (Abbey-Lee et al., 2016; Kleindorfer et al., 2013a; Kleindorfer, Fessl & Hoi, 2005; McQueen et al., 2017; Williamson & Fagan, 2017; Zuberbühler, 2001). Many previous studies assessing avian predator response used physical models to simulate a predator (Colombelli-Négrel, Robertson & Kleindorfer, 2010a; Krams et al., 2014; Magrath, Pitcher & Gardner, 2009; Mutzel et al., 2013). As described previously, we cannot discount that a visual predator stimulus might have evoked a different, and perhaps stronger, defensive response from our subjects, and potentially displayed a clearer relationship between proactive personalities and playback response. Because most avian predators do not call while hunting, our predator playback may have been interpreted to mean that a predator was present in the area but did not necessarily pose an immediate threat (see also Abbey-Lee et al., 2016). Even so, the use of audio playback does have potential benefits. Model presentations, while effective for eliciting alarm responses and defensive behaviours, also require the bird to have a clear line of sight to the model, which is often impractical in the dense vegetation in which many fairy-wrens build their nests (Magrath, Pitcher & Gardner, 2007; Magrath, Pitcher & Gardner, 2009).

Contrary to our predictions, sex did not predict an individual’s playback response, despite most males being in their nuptial plumage at the time of testing. Highly conspicuous individuals tend to be risk-averse in the presence of predators, choosing areas of higher cover when calling for danger (Katz et al., 2015; Ximenes & Gawryszewski, 2020). Our results may be explained by the dense vegetation of our study area, where even highly conspicuous males were able to respond to the predator playback without significantly increasing their risk compared to their counterparts. A future investigation may account for this by considering the vegetation cover of the area surrounding the simulated predator. We also cannot exclude that males perceived the lower risk posed by the playback (compared to a visually present predator) and used the opportunity to display their potential as mates, as found by Langmore & Mulder (1992) in superb fairy-wrens and by Greig & Pruett-Jones (2008) in splendid fairy-wrens (Malurus splendens). Future experiments with model presentations may be able to demonstrate this.

Contrary again to our predictions, an individual’s response to predator playback did not correlate with its exploration behaviour in a novel environment (measured in captivity) nor its response to human handling (boldness measured in captivity). This contrasts with previous work in great tits (Parus major) showing associations between a bird’s exploration behaviour in captivity and its response to risky situations in the wild. Specifically, more exploratory great tits were more responsive to playback simulating a conspecific intruder (Amy et al., 2010), were less neophobic towards a novel object at their nest (Cole & Quinn, 2014), and called more in response to a nest intruder (Hollander et al., 2008). In superb fairy-wrens, it may be the case that exploration in a controlled captive environment simply does not correlate with behaviour in the wild, as has been reported in zebra finches (Taeniopygia guttata) (McCowan et al., 2015). However, the negative relationship between fairy-wren exploration behaviour and apparent survival (see Hall et al., 2015) implies that personality traits measured in captivity correlate at least with some aspect of their wild behaviour. Clearly, further work is needed to determine how fast- and slow-exploring individuals differ in their behaviour in the field and the fitness consequences associated with each behavioural phenotype. Similarly, an individual’s boldness in captivity (struggle response to human handling) did not predict its behaviour in the field. Our handling assays were intended to simulate the experience of being seized by a predator and having to escape. Because of this, the fitness benefits associated with a strong or weak handling response may only be apparent during close physical altercations with a predator, and not necessarily during the earlier stages of a predator encounter. In a study of blue tits (Cyanistes caeruleus), Fresneau, Kluen & Brommer (2014) found a sex-specific relationship between handling response and nest defence, with less responsive females defending their nestlings more vigorously. Notably, nest defence was measured while experimenters removed a nestling from the nest, simulating a nest predator more directly and intensely than the vocal playback used in our study. Therefore, the potential fitness implications of a fairy-wren’s response to handling still need to be tested in scenarios in which predators are perceived to pose a direct and immediate threat.

Conclusion

This study adds to the expanding literature of how individual personality may correlate with behavioural responses to predators (Dingemanse & Réale, 2005; Hollander et al., 2008; Krams et al., 2014). Of the three personality traits we measured in captivity, only one (aggressiveness, measured in a mirror stimulation test) significantly predicted that bird’s behavioural response to a simulated predator in the wild. Our study suggests that overarching behavioural syndromes may not be as static as previously suspected (Hall et al., 2015; Wuerz & Krüger, 2015), with specific personality traits predicting situational behavioural responses. With only one of three traits predicting wild behaviours, it is likely that researchers’ choice of personality trait helps determine whether they find significant relationships between personality and other wild behaviours. Hence, in future investigations of this kind, we suggest it may be useful, or even essential, to measure multiple personality traits. Future studies should investigate this relationship between multiple populations and examine how risk-taking behaviours correlate with actual mortality and breeding/foraging success.

Supplemental Information

Supplemental Information 1 Raw data

Response of superb fairy-wrens at Cleland Wildlife Park to the playback of currawong or willie wagtail calls.

Click here for additional data file.

Supplemental Information 2 Output from GLMM testing whether personality traits measured during short-term captivity (boldness, exploration, and aggressiveness) correlated with behaviours in the wild in response to a simulated local predator (PC_Playback)

Model included (1) boldness (PC_Handling), (2) exploration (PC_Exploration), (3) aggressiveness (‘mirror attacks’, as a quadratic variable), (4) sex (male, female), (5) playback order (whether the predator playback was conducted first or second), and (6) number of responders (the number of group members that responded to the playback) as fixed factors and ‘Territory ID’ as a random effect. Statistically significant (≤ 0.05) values are marked in bold (N = 40 individuals).

Click here for additional data file.

We thank Cleland Wildlife Park for providing access to our study site. We also thank Christine Evans, Lauren Common, and Connor Panozzo for help with fieldwork and data collection. Finally, we thank the reviewers for their helpful comments on the manuscript.

Additional Information and Declarations

Competing Interests

Author Contributions

Animal Ethics

Field Study Permissions

Data Availability

The authors declare there are no competing interests.

Jack Bilby conceived and designed the experiments, performed the experiments, analyzed the data, authored or reviewed drafts of the article, and approved the final draft.

Diane Colombelli-Négrel conceived and designed the experiments, performed the experiments, analyzed the data, prepared figures and/or tables, authored or reviewed drafts of the article, and approved the final draft.

Andrew C. Katsis conceived and designed the experiments, performed the experiments, analyzed the data, prepared figures and/or tables, authored or reviewed drafts of the article, and approved the final draft.

Sonia Kleindorfer conceived and designed the experiments, authored or reviewed drafts of the article, and approved the final draft.

The following information was supplied relating to ethical approvals (i.e., approving body and any reference numbers):

The Flinders University Animal Welfare Committee approved this study (permit E480-19).

The following information was supplied relating to field study approvals (i.e., approving body and any reference numbers):

We conducted fieldwork at Cleland Wildlife Park under permit Z24699-17 (approved by the Department of Environment and Water)

The following information was supplied regarding data availability:

The data are available in the Supplementary Files.

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
