# Peer review of "When aggressiveness could be too risky: linking personality traits and predator response in superb fairy-wrens"

_PeerJ, doi:10.7717/peerj.14011_

## Round 0.1 · original submission · Major Revisions

Dear authors,

Thank you for your submission to PeerJ. Based on the comments from three expert reviewers, this paper will require Major Revisions to be considered for publication in PeerJ. Specifically all three reviewers raise methodological concerns, requiring many reanalyses, as well as concerns regarding the conclusions based on the results. Please carefully go through each review and address the major points brought up.

Please submit a response to reviewer comments, tracked changes version of your manuscript, as well as a clean version of your manuscript upon resubmission.

Please do not hesitate to contact me if you have any questions.

Brandon P. Hedrick, Ph.D.

Reviewer 1 ·

Basic reporting

Language is clear – however there are places where the word choice or order makes the meaning difficult to follow. Recommendation provided.
Perhaps b/c the results are different than predicted – the authors are missing a bit of literature that specifically links con-specific aggression and predator aggression/defence. Some literature is missing and others are cited strangely. But overall sufficient.
Structure is fine. There is only 2 figures – 1 for the results, and it is not referenced in the MS. The model results are provided in supplemental materials. So it’s hard to assess the GLMM models
Raw data is easy to follow and clear.

Experimental design

The question is interesting, the data sufficient to answer the question. The question is interesting and the methods sufficiently detailed for repeating. No ethical concerns.
The subject matter is appropriate for the Aims and Scope.

Validity of the findings

I have two issues with the analyses –
1. Order of trials (1/2) is used as a random variable or excluded. This doesn’t follow best practice.
2. The authors decide to divide behavioural responses to the predator into two seperate PCAs according to what makes sense to them. I think it would be stronger to include them all in one PCA and then examine the results PC scores - being lead by the data - instead of making such decisions a priori.

I am also unclear as to why the author do not fit a model with all the personality metrics to see if accounting for other behaviours better explains response.

The authors conclude that their results support their hypothesis – but this is a stretch. Of five personality metrics, only one is related to anti-predator responses, and that variable was a binomial.

Additional comments

This is an interesting questions and the studies are well done. It is a valuable contribution to the field of animal behaviour and adds to the growing literature that illustrated there in important, consistent individual variation in behaviour across contexts.
It is a lot of work to collect this much behavioural data on free-living birds. I commend to authors on the body of work.

I have some suggestion to improve the MS.
1. I think you overstate the results by concluding that personality predicts response to predator. First, because predator response is an aspect of personality itself. Second b/c the only variable related to response was a binomial measure (of 5). You also miss an opportunity to discuss some interesting relationships between different forms of aggression – and the associated body of work.
2. There are some basic gaps – you never refer to the figures, you discuss color and sex in the intro and discussion but those are missing from the results. The GLMM model results are not presented in the results, and Order should not be a random effect (detail below).
3. There are some parts of the intro where the language implies causation when the study is correlational and other places where there are big logical jumps or assumptions that I’ve tried to point out.

General comments
You never refer to any of the figures, and I would suggest more figures – perhaps correlations matrixes between all the PC scores? Or scatterplots?

A fair amount of the intro discusses colour and sex differences but there is nothing in the results about either?


Abstract – general
Many readers won’t be familiar with Australian birds and there is nothing in the abstract mentioning that these are birds if you don’t know them by name. I suggest adding some descriptors to the species names e.g. a common Australian passerine

L12 – This sentences don’t really follow the previous sentence – rather than illustrating how proactive behaviour can improve survival , you stated that it is likely to reduce survival – perhaps reword to make it clearer that resource acquisition can increase survival/fitness in some cases.

L15 – some would say that anti-predator behaviour is part of personality – so reword to make it clear that you are testing to see if some axes of personality are related to another – or whether stress/exploration/aggression is related to predator response to be more specific.
Further – the in-hand struggle and mirror test are not really standard – they are relatively new metrics, did you mean standardised?

L18 – suggested revision - Superb fairy-wrens showed a significantly stronger response to the predator playback…..

L22 – I think you mean was unrelated to playback response – otherwise you are suggesting that the novel environment test was a treatment that didn’t affect predator response.


Intro-
L51 – highly active does not mean more efficient. They could be running around crazy but not getting much food. And again L53 it is not efficiency – which would mean you get the same about of resources with less work/time – but activity.

L 58 – weird to have the citation in the middle of the sentence – or to have no citation for the second half of sentence.

L63 – it has also been suggested that females have higher energy needs – so are less careful – see Cain et al. 2019

L67- again influenced implies and experimental treatment (for example- did exogenous testosterone influence behaviour) – you are examining correlations/associations so be careful with invoking directionality/causality.
Also – as in the abstract you should mention that these are songbirds – many readers won’t know what a fairy-wren or currawong is.

L79 – do you know that these behaviours are maladaptive or are you assuming? It is common for birds to mob predators and it is usually assumed to be adaptive. – though I agree it is riskier – risky does not mean maladaptive – it depends on the overall effect on nest survival/predation etc.

L83 – why do you specify dominant females here? You haven’t mentioned the cooperative nature of SFW yet, and there are rarely more than one female in a SFW breeding group.

Methods –

L94 – I recommend mentioning this is during Austral summer/ breeding season.

L99 – what happens to the females?

L104 – if this population has been monitored since 2010 why not use predation rates here instead of Mount Lofty?

L114 – is it an alarm song or an alarm call? L183 it is a call

L125 – because of the order, this makes it sounds like you are considering morphological measures as a behavioural response. It is clearer after reading the next sentences that you counted struggle during measurement – but I think you could rearrange, and just say we also measured struggled while taking 5 morphological measures.

L157 – hasn’t repeatability in this species already been tested in a different population? The Melbourne birds? Should be cited as well as your own data.

L164 – how many different calls were used? How many times were each used?

L176 – did you make sure the birds were present? Is it possible that you started PB when birds were away foraging?

L186 – This seems strange – no metrics of vigilance or time to return to nest or time to leave area etc? Better to just say you didn’t analyse that behaviour than no behaviour were observed there is always behaviour even if it is just roosting or siting motionless – that is a behaviour.

L204 – why use this approach instead of including all 4 variables in one PCA? In theory then you would wind up with ~2 PC scores that each load on different important sources of variation – that might wind up with investigation and risk being PC 1 and 2, or it might not. But by splitting them up you miss the opportunity to see if those behavioural variable are related. If they are that tells you that they are different aspects of the same behaviour/personality trait, if they aren’t then they aren’t…
I don’t see the utility of deciding a priori that they are different and should be separated. You should let the data lead you.

L217 – why not include all 3 variables in one model? Sex x behaviour interactions?

L221/223 – Order should not be a random effect – it should be a factor. It is part of the experiment and there are only 2 levels. Generally random effects should be more numerous. See http://bbolker.github.io/mixedmodels-misc/glmmFAQ.html
“One point of particular relevance to 'modern' mixed model estimation (rather than 'classical' method-of-moments estimation) is that, for practical purposes, there must be a reasonable number of random-effects levels (e.g. blocks) — more than 5 or 6 at a minimum.”


Results –
L226-32 – In addition to the statistical results, it would be nice to have some real life numbers here. How much faster was the response to the predator? How much more time did they spend in 5m? The stats tell us the results are likely not due to chance – but we still need to see the actual results.

L235 – folks have different opinions on this – but a P=0.07 is a fairly strong effect for behaviour and the sample size. I would suggest it is worth saying there was a relationships, but didn’t reach the alpha cut-off. See this article
https://www.sciencedirect.com/science/article/pii/S0169534721002846

L237 – aggression was binomial correct? I think clearer to reword. E.g. animals that attacked the mirror have stronger response’s to the playback, approached faster and closer.

Nothing about sex or colour?


I would prefer Table S1 in the main article – This is critical to evaluating the results. I’d also prefer to see the estimate instead of F values and dfs. This table is pretty rough and could be made easier to follow.

Discussion –
L247 - I find this summary/framing of the results a bit misleading. You measured a number of aspects of personality, and the only thing that predicted anti-predator aggression is conspecific aggression. So did you really find that personality predicts predator response, or did you find that aggression in one context is related to aggression in another context?
Plus the measure was a binomial – this really over states the strength of the result.

Given the results you are really missing the following papers

• Witsenburg, F., Schu ̈rch, R., Otti, O. & Heg, D. 2010: Behavioural types and ecological effects in a natural population of the cooperative cichlid Neolamprologus pulcher. Anim. Behav. 80, 757—767.
• Huntingford, F. 1976: The relationship between anti- predator behaviour and aggression among conspecifics in the three-spined stickleback, Gasterosteus aculeatus. Anim. Behav. 24, 245—260.
• Cain, KE, MR Rich, KA Ainsworth & ED Ketterson. 2011. Two sides of the same coin? Consistency in aggression to conspecifics and predators in a female songbird. Ethology, 117 (9): 786–795

L262 – most predators use surprise – risks may not be high once the predator has been seen.

L271 – there are papers on this in fairy-wrens, but no citations?

L294 – Neither Cain or Medina have behaviour in the studies – not sure why they are referenced here.

L301 and in SFW – Langmore
Langmore, Naomi E. and Raoul A. Mulder. “A Novel Context for Bird Song: Predator Calls Prompt Male Singing in the Kleptogamous Superb Fairy‐wren, Malurus cyaneus.” Ethology 90 (2010): 143-153.

Reviewer 2 ·

Basic reporting

Paper is generally very well written.

Literature is good; however, a very relevant and important paper (Teunissen et al 2021 Current Biology) is missing, which draws attention to a potential shortfall of the current study.

Raw data is shared. Structure is good. Figures / Tables are fine.

Results are relevant to hypotheses; however, I do have a problem with the variables considered important (and unimportant) - see below.

Experimental design

Recent work by Teunissen et al. (2021 Current Biology) highlights the importance of social context in nest defence behaviours in fairy-wrens. In this current study, the response to the treatment playbacks was conducted on a natural group of fairy-wrens, and then the behaviour of individuals were analysed. This is potentially problematic, because if social context can affect individual behaviour, this should then be considered in the analysis. The authors state that “… ‘Territory’ (to account for variation among groups) [was included] as a random effect”; however, this is a bit concerning because having looked at the raw data (which the authors have included) this seems to refer simply to group ID (i.e. birds were used from 14 groups) rather than including any information about variation among groups (i.e. group size, or the number of birds that attended a particular trial). I must admit that I did feel a little deceived when seeing what this referred to in the raw data.

Given this, I find the manuscript hard to review, as I believe that a major do-over of the analyses is required (i.e. include metrics of group composition, especially during a given playback) before a more detailed review can be completed. Outside of this major point, I think the manuscript was generally well-executed (i.e. well-written and the logic generally flowed well). It is just difficult to properly judge the statistics, and therefore results, as they currently stand. Furthermore, I do think that the decision to analyse individual behaviour from a group trial is a short fall in the design of the study, and therefore a discussion of the potential consequences of this should also be included in the discussion (because currently there is none).

Validity of the findings

See 2, above. I have concerns with the approach taken (extracting individual behaviour from a group trial and then not discussing the importance of the social group influence on the behaviour of the individual).

Additional comments

Data availability statement should be changed to note that data is available in the supplementary materials.
L23: Is “influence” the right word here, or would something like “correlate” be more appropriate?
L 42: Probably worth introducing the proactive reactive axis.
L 85-87: Should nest your predictions within relevant work (e.g. McQueen et al. 2017).
L 222: Did you analyze helper and dominant individuals separately? This does not seem to be clear.
L 229: “territory” appears (looking at the raw data) to refer to group id? Is this correct? If so, the text is a bit deceptive.
L 281-283: References needed to back this up
L 285: Did Hall et al. say lower survivorship? Or, lower re-sighting rates in the following year?

Reviewer 3 ·

Basic reporting

The ms is generally clearly written, sensibly structured, and with good reference to previous work. It tests specific hypotheses that are potentially of broad interest, and gathers relevant data.

There could be more background on what constitutes "personality". All definitions require that measures of "personality" are consistent over time within an individual. Some require that personality is a syndrome of related attributes (eg bold, exploratory, aggressive vs shy, retiring, non-aggressive), rather than referring at any one trait. Clarifying definitions is important, as only one specific measure of "personality" (response to a mirror) appears to show a "significant" association with response to predator playback, and there is only weak evidence that it is repeatable within an individual (marginally "non-significant"). Other specific measures did not influence response to predator playback, yet were more repeatable. It is necessary to have a clear understanding of of "personality" to interpret results and draw conclusions.

Experimental design

The research question is reasonably well defined, but the methods lack detail and are problematic. One of the key statistical analyses appears to be invalid.

1) There is insufficient reporting on playback tracks. a) Give XenoCanto reference numbers in the text or in an appendix. b) What type of currawong call was used? c) How many exemplars of each type of playback were used, and from how many individuals? The text in the methods implies one call; the statistical methods imply more than one. d) In the 3 mins of playback, how many calls from how many individuals were used? e) Comment: yes, you can save wav files after opening XC files, but XC files are compressed sounds (mp3), so the quality is reduced even if you save them in wav format. f) What device, settings and specific measures did you use when measuring the amplitude of playbacks? g) What is the evidence that these amplitudes are similar to the natural amplitudes? f) What does playback ~ 90 dB mean; that is, how much variation was there about the mean, and were currawong and wagtail sounds the same amplitude?

2) Responses to playback. a) Responses by individuals within a group are likely to be affected by what others do, and so are almost certainly interactive. Data from the different individuals within a group therefore cannot be considered independent. Fitting the random term "Territory" does not solve the problem, because it effectively just adds a constant to account for group effects statistically, but assumes that responses of individuals within the group are independent. b) Playback treatment (predator vs control) was not blind to the people scoring the behaviour. This deserves comment. Why wasn't video used to get around this problem? Was audio recording used to quantify the number of calls? c) What types of "alarm calls" were given by focal birds (eg mobbing, aerial, alarm song)? Did they give trill song in response to currawong sounds, which apparently is a natural context for these calls. d) Measures of response to control vs predator treatments need to be given.

3) Analyses. a) As noted, GLMMs with Territory as a random term do not sole the problem that individuals within a group are likely to be affected by how others in the group respond. b) It's unclear why there are 2 PCs to measure response. What's the rationale, for example, of having time with 5m but not closest approach in the PC for "Risk"? c) Personality vs response analyses used a normal distribution (so had identity link), yet Figure 2 shows that residuals would not be normal.

Validity of the findings

Given problems with methods and statistical analyses listed above, it does not appear possible to draw robust conclusions. In addition, 2/3 measures of "personality" found no effect on response, including "exploration", which a previous study suggested was important. So even if results and analyses are taken at face value, the conclusions are less clear than suggested.

Overall, the question posed is interesting and the results are suggestive that further work would be worthwhile and interesting. The current study, however, is problematic in design and interpretation.

---

## Round 0.2 · Minor Revisions

Dear authors,

Thank you for a careful revision and for responding to the previous round of reviewer comments. I quite enjoyed reading this paper. There are just a few minor points that the reviewers bring up that will increase the clarity of your paper. Adding in some more reasoning behind methodological reasoning is necessary in addition to a few minor word changes here and there, after which I believe the paper will be publishable in PeerJ.

In addition to reviewer comments, I had the two following notes:

Grammar issue in Table 2 description. Probably ‘time near the speaker’?

Figure 3/4. Just double checking the horizontal line is the mean and not the median. Odd you'd have the mean and quartiles in a boxplot.

When you submit your revision, please include a response to reviewer comments document, a tracked changes version of your manuscript, and a clean version of your manuscript.

Thank you for your submission. Please let me know if you have any other questions.

Best,

Brandon P. Hedrick, Ph.D.

Reviewer 1 ·

Basic reporting

I commend the authors on a solid revision. The paper is much improved. I have a few minor comments for consideration.


In the abstract - There is a general sense that proactive responses are inherently costly/have negative consequences for fitness. But effective predator deterrence would be hugely beneficial. It is just a trade-off increasing personal risk but increasing offspring survival, rather than just being risky. This is more nuanced in the introduction and I recommend that more tempered language be extended into the abstract.

Experimental design

The research is interesting, the data sufficient to answer the question, and the methods sufficiently detailed for repeating. No ethical concerns.
The subject matter is appropriate for the Aims and Scope.

Validity of the findings

Analyses are improved but I find it strange to consider a binary response as a personality trait. In theory the mirror test would show lots more variation- time close to the mirror, perch changes, vocalizations etc. Further, these test are so difficult to do and birds show so much variation – it seems waste for it to be reduced to y/n

Additional comments

Line 49 – some parentheses missing here or too many??
L 102 responding more slowly?? Or quickly if fast-exploring?
L132 –confused about the two different predation rates. If Cleland is in Mt Lofty – what’s the difference between the two?

L275 – in the methods you say you do not count alarm calls but here you say there was no difference in the number of alarm calls?
L280-3 how do you correlate a categorical variable (mirror y/n) with a continuous variable? Wouldn’t that be a Mann-Whitney U?
L290 are these models binary GLMMs?

·

Basic reporting

Your paper is written with very unambiguous and professional English language with generally a well thought out structure. I think your methods, results, discussion and conclusion are very enjoyable to read and I have no comments for improvement there. With respect to your introduction, I think it might benefit by getting straight to the point. I appreciate that your first paragraph is a very nice introduction to animal personality more generally, but I feel the depth that you go into is unnecessary. It is good to state what animal personality is, considering the novelty of this field, however I think your first sentence is a bit vague and not all that exciting. I would be tempted to go straight to a comment similar to the first sentence of your discussion, perhaps modified slightly.

I feel like your 2nd paragraph very nicely explains what we know about personality traits so far, therefore it would serve as a nice follow on from an opening statement in a first paragraph. Especially since the 2nd paragraph ends with one of the major gaps. I think your introduction would benefit by combining the last sentence of your first paragraph with the last sentence of your 2nd paragraph, I felt like there was an element of repetition that could be minimised and would make the area of unknown research stand out more. You refer to table 1 and figure 1 very appropriately and timely in your introduction, preventing the reader from trying to create their own image of how everything looks in their head – you’ve laid it out for them. Perhaps by labelling the types of personality axes in table 1 (e.g. a) shyness-boldness, b) exploration-avoidance…) you can then refer to them specifically (a, b..) in your text when you refer to them so we can follow you more easily. I really like the rest of your introduction, supported very nicely by relevant studies and sets up your aims and study very well.

The raw data you provided is clear and easy to follow.

Experimental design

Your aims and methods are very well defined, but I was curious about your decision to perform the playback during the incubation phase. Since you were interested in controlling for sex differences is investment and conspicuousness, surely at the incubation stage there would be a greater difference between the sexes in investment in the nest as opposed to the nestling stage? As you said in lines 80-81: “when early parental investment is higher for female birds (through the cost of laying and incubation), mothers may tolerate higher levels of risk when defending their investment”. If you had carried out the playback during the nestling stage all/most individuals in the group would be investing in feeding. You could even quantify the investment of individuals then, by looking at provisioning rates. Therefore, I think it would be good if you gave a bit more information on why you chose the incubation stage for your wild experiment.

Also, would you wait for the female to be on an off-bout before playing the calls? It might be good to specify this too, since it could definitely affect how the female responds. I know often that incubating females may sit tight until a threat is directly next to the nest (e.g. when you walk past and flush an incubating female).

I was a bit confused about your field sites. You mention only one in your study site and species description, however in lines 132-133 you talk about Mount Lofty Ranges and Cleland Wildlife Park as if they are two different sites with different nest predation rates. Perhaps clarify this a bit better, or if indeed you do have two sites, you need to discuss this and the differences in nest predation rates between the two areas, maybe even consider including it as a variable in your statistical analysis.

Do you think there could be an effect of banding unbanded individuals on your personality tests? I think it would be good to comment on this in your methods. Did you band birds before their personality tests? If so how long before hand? Also, did you consider the differences in response you would get from individuals that had been caught and handled before compared to those that were unbanded and had never been handled?

Validity of the findings

I think you should say why you did not include status (helpers or dominants) of individuals in your analysis, I would have though this would have an influence on their response, both in the personality tests and the playback. If anything, they will have different investment incentives at the nest, never mind that dominants are likely to be more aggressive and bold than helpers.

I think your findings are discussed in good detail, however I would have appreciated it if you had made it a bit more obvious how your research fills your knowledge gap, especially in the abstract. I think by the discussion you have made the point very nicely, but I did not get what your research had done for the field in your abstract.

You explain the context very nicely in your abstract, but I think you could add a link in there that tells us what is missing in your field and why you are doing the study. I don’t immediately get that there is an urgent need for your research until I read your introduction.

Additional comments

I would like to congratulate the authors on a generally well written and exciting piece of work. Your research overlaps very nicely with my area of expertise, therefore most of my comments will focus on your experimental design, or your description of it, with some comments about how you communicate the theoretical background.

What I took away from your paper is that you addressed essentially 2 gaps in the field of personality: 1) that links between personality measured in captivity and wild behaviours (i.e. different contexts) are understudied, and 2) the influence of other factors that effect risk-taking behaviour (such as sex and conspicuousness) are rarely considered. Therefore, you firstly assessed 3 personality traits of captured individual superb fairy-wrens: 1) boldness, via human handling, 2) exploration, via novel environment test (cage), and 3) aggressiveness, using a mirror stimulation test. You then performed predator playbacks at incubating nests with group members of known personality phenotype. The strength of their response to the playback was related to their personality phenotype. You found that none of the personality traits were correlated, but that more aggressive individuals tended to be more explorative. You also found that more aggressive individuals during the mirror stimulation test, respond stronger to the predator playback and that this indicates specific personality traits instead of a correlated suite of traits are likely to predict context specific wild behaviours.

---

## Round 0.3 · accepted · Accept

Dear authors,

Thank you for following reviewer comments closely and your changes to the manuscript. I now find it to be publishable in PeerJ and am moving it forward to ‘accept’. There are a few very minor things that need to be corrected prior to the manuscript being published (see below).

Line numbers are based on the tracked changes version

Line 23: Rhipidura leucophrys spelling

Line 138: Ambystoma barbouri spelling.

Line 432: Your r^2 is negative? Is this not just the correlation coefficient? Double check this since squaring it makes it positive.

Remember to thank the reviewers in your acknowledgements.

Thank you for your submission to PeerJ. Please contact me if you have any questions.

Best,

Brandon P. Hedrick, Ph.D.